# Effects of Supplementation with a Quebracho Tannin Product as an Alternative to Antibiotics on Growth Performance, Diarrhea, and Overall Health in Early-Weaned Piglets

**DOI:** 10.3390/ani11113316

**Published:** 2021-11-19

**Authors:** Min Ma, James K. Chambers, Kazuyuki Uchida, Masanori Ikeda, Makiko Watanabe, Yuki Goda, Daisuke Yamanaka, Shin-Ichiro Takahashi, Masayoshi Kuwahara, Junyou Li

**Affiliations:** 1Animal Resource Science Center, Graduate School of Agricultural and Life Sciences, The University of Tokyo, Kasama 3190206, Japan; muchsojina@outlook.com (M.M.); aikeda@g.ecc.u-tokyo.ac.jp (M.I.); awatanabe.makiko@mail.ecc.u-tokyo.ac.jp (M.W.); 2Veterinary Pathophysiology and Animal Health, Graduate School of Agriculture and Life Science, The University of Tokyo, Bunkyo-ku, Tokyo 1138657, Japan; akuwam@g.ecc.u-tokyo.ac.jp; 3Laboratory of Veterinary Pathology, Graduate School of Agriculture and Life Science, The University of Tokyo, Bunkyo-ku, Tokyo 1138657, Japan; achamber@mail.ecc.u-tokyo.ac.jp (J.K.C.); auchidak@mail.ecc.u-tokyo.ac.jp (K.U.); 4Laboratory of Cell Regulation, Graduate School of Agriculture and Life Science, The University of Tokyo, Bunkyo-ku, Tokyo 1138657, Japan; leonardo.fibonacci11235813@docomo.ne.jp (Y.G.); atkshin@mail.ecc.u-tokyo.ac.jp (S.-I.T.); 5Laboratory of Food and Physiological Models, Graduate School of Agriculture and Life Science, The University of Tokyo, Kasama 3190206, Japan; adyama@mail.ecc.u-tokyo.ac.jp

**Keywords:** quebracho tannin, weaned piglets, diarrhea, growth performance, intestinal morphology

## Abstract

**Simple Summary:**

The restriction of the use of antibiotics in swine production worldwide has influenced pork production efficiency. New in-feed additives must be sustainable, prevent diarrhea in early weaning piglets, and promote growth performance. Novel in-feed additives, probiotics, prebiotics, organic compounds, mineral salts and vegetable extract have been extensively studied; most have shown some limitations that discourage extensive use. We investigated the plant extract MGM-P (a quebracho tannin product) as an alternative animal feed additive to antibiotics. We considered its unique structure, antibacterial, antioxidant, radical scavenging, and anti-inflammatory activities, and sustainability. We began with a low-level addition trial; 0.3% MGM-P had a more robust effect than 0.2% MGM-P. The findings demonstrated that 0.3% MGM-P supplementation prevented diarrhea in 21-day-old weaned piglets, improving piglet health without adversely influencing growth performance. Practical studies of the mechanisms underlying the effects of MGM-P and the optimal amount for supplementation are needed to confirm our findings.

**Abstract:**

This study assessed the feasibility of using a vegetable extract, MGM-P (quebracho tannin product), as an alternative to antibiotics for weaned piglets; it investigated MGM-P effects on growth performance, diarrhea, and overall health in early-weaned piglets. In total, 24 piglets were allocated to three treatment groups fed basal diets supplemented with 0, 0.2%, or 0.3% MGM-P for 20 days. The addition of 0.3% MGM-P to the diet of early-weaned piglets improved diarrhea incidence, hematological parameters, and intestinal mucosa structure. Furthermore, the addition of 0.2% or 0.3% MGM-P to the diet of early-weaned piglets did not affect their overall health. Importantly, MGM-P had no effects on average daily gain (ADG), average daily feed intake (ADFI), or feed conversion ratio (FCR). Gut morphology analysis showed that treatment with 0.3% MGM-P enhanced the jejunal villus height (*p* < 0.05) while reducing the ileal crypt depth (*p* < 0.05) and colon mucosal thickness (*p* < 0.05). Collectively, the findings suggested that the use of MGM-P as an alternative to dietary antibiotics could improve diarrhea incidence in early-weaned piglets without negative effects on growth performance or overall health.

## 1. Introduction

The modern swine industry weans piglets at the ages of 19–25 days to increase sow reproductive efficiency, thus improving annual productivity [1]. However, this early weaning poses substantial physiological, environmental, and social challenges for piglets, including abrupt separation from their mothers, exposure to unfamiliar piglets, establishment of a new social hierarchy, different housing conditions, and changes in feed sources [1,2]. Piglets under weaning stress are at risk of severe diarrhea, growth reduction, and even death. Diarrhea is a particular problem for early-weaned piglets. Artificially reared early-weaned piglets are protected from enteropathogens through dietary supplementation of additives such as antibiotics. However, after more than 70 years of such antibiotic supplementation, there is global concern that antimicrobial resistance has made human medicines less effective [1]. Most developed countries have begun to ban the use of antimicrobial agents as feed additives. Thus, there is an urgent need to ensure high pork productivity and identify sustainable alternatives to antibiotics.

Tannins are widely used in traditional human medicine to fight chronic diarrhea because of their abilities to prevent intestinal bacteria and parasites [3,4]. The unique structures and mechanisms of tannins provide beneficial effects in pig farming, in relation to their antimicrobial, antioxidant and radical scavenging, and anti-inflammatory activities [5]. Thus, tannins offer alternatives to antibiotics for the prevention of diarrhea in early-weaned piglets.

Tannins are astringent polyphenols; they are classified as hydrolyzable or condensed. Tannins have been presumed to constitute anti-nutritional substances because they can precipitate proteins, inhibit digestive enzymes, and reduce nutrient use [6]. However, the so-called “anti-nutrition effect” of tannins can in fact protect rumen-bypass protein and improve protein metabolism in ruminants [7]. In recent years, there has been a gradual increase in research concerning the use of tannins in monogastric animals, although analysis of the anti-nutrition effect remains challenging. Jamroz et al. [8] added chestnut tannin, a hydrolyzable type of tannin, to the feed of newborn chickens for 42 days; they found that the addition of 250 or 500 mg/kg chestnut tannin had no effect on chicken growth, whereas the addition of 1000 mg/kg chestnut tannin reduced the final body weight (BW). Antongiovanni et al. [9] added chestnut tannin to the diet of growing pigs; they found that the addition of 0.5% chestnut tannin reduced the digestion of both dry matter and nitrogen. Although Biagi et al. [10] found that supplementation of fodder with 4.5 g/kg chestnut tannin improved the feed conversion ratio (FCR) of weaned (28-day-old) piglets from 1.32 to 1.39, they found no difference on average daily gain (ADG) or final BW. Compared to hydrolyzed tannins, condensed tannins often exhibit stronger antibacterial activity [11]. Quebracho tannin, extracted from the heartwood of the quebracho tree (*Schinopsis lorentzii*), is a representative condensed tannin. To our knowledge, there are very few data regarding the use of quebracho tannin for post-weaned piglets. Hydrolyzable tannins are hydrolyzed by weak bases, weak acids or weak enzymes to produce carbohydrates and phenolic, gallic and ellagic acids. Thus, hydrolyzable tannins are readily affected by basal diet composition. The hydrolyzed phenolic, gallic acid, and ellagic acids, have potential antibacterial effects, which may to lead to variable research results. Condensed tannins have a stable structure and are not hydrolyzed. Condensed tannins bind to and precipitate proteins and various other organic compounds (e.g., amino acids and alkaloids), supporting their addition to the diets of weaned piglets. Caprarulo et al. [12] added a 1.25% quebracho tannin and chestnut tannin mixture to the diet of weaned piglets; it did not affect piglet growth performance, although the diarrhea incidence in the first 14 days post-weaning was higher in the tannin group (5.00%) than the control group (3.39%). Su et al. [13] reported that the addition of 0.1% quebracho tannin to 15-kg BW piglets significantly reduced feed intake and BW gain during the first week; both aspects recovered during the following week. The results of studies conducted during the weaning, growing, and finishing phases have thus shown heterogeneous responses, which may be related to the amount of tannin, type used, age of animals, ingredients of the basal diet, and hygiene and storage status. Notably, tannins are present in various feedstuffs and ingredients for animal diets (e.g., corn, wheat, and barley). Thus, tannin-rich feedstuffs could increase the concentration of dietary tannins [14]. The present study aimed to evaluate the effects of adding a 0.2% or 0.3% quebracho tannin product (MGM-P) to the commercial feed diet of 21-day-old weaned piglets in terms of growth performance, diarrhea, and overall health.

## 2. Materials and Methods

### 2.1. Materials

Commercial MGM-P was provided by Kawamura Ltd. (Tokyo, Japan); condensed tannins comprised more than 50% of the overall extract (Table 1).

### 2.2. Animals, Treatments, and Experimental Design

The experiment was conducted at the Animal Resource Science Center of the University of Tokyo (Kasama, Japan), and approved (P20-096) by the Animal Care and Use Committee of the Faculty of Agriculture, The University of Tokyo. Three pregnant specific-pathogen-free sows were purchased from Nakamura Chikusan (Ibaraki, Japan) at 1 week before delivery. Thirty piglets (Duroc × Landrace × Yorkshire) were born within 2 days. The lactation period was 21 days. The male piglets were castrated at 1 week of age. Concurrently, all piglets were numbered; from 2 weeks of age, they were provided Antibacterial-substance-free (ASF) early-stage fodder (Marubeni Nisshin Feed, Tokyo, Japan) as creep feed, and during the experimental period. The piglets were weighed at 21 days of age. Using the Experimental Animal Allotment Program (version 1.1) in accordance with the method established by Kim and Lindemann [15], 24 piglets were selected and divided into three groups (*n* = 8 per group) according to weight and sex. The animals in each group were divided into two identical pens containing four piglets each (Table 2).

The control group received ASF early-stage fodder without any added MGM-P; the 0.2 MGM (0.2% MGM-P) and 0.3 MGM (0.3% MGM-P) groups received ASF early-stage fodder with 2 g/kg and 3 g/kg MGM-P, respectively. The experimental period was 20 days after weaning.

### 2.3. Diet and Animal Management

Table 3 lists the ingredients in ASF early-stage fodder, which meets the National Research Council standards (Table 4) [16]. All piglets were raised in the same high-bed nursery house, equipped with an air conditioner and mechanical ventilation, molded plastic pen floors, a feed hopper, and a SUEVIA water cup. Throughout the experimental period, piglet health status was checked and recorded twice daily. 

### 2.4. Growth Performance

The piglets were weighed at the same time on days 1 (weaning day), 7, 14, and 20 after weaning; the amounts of fodder consumed in each pen were recorded. The ADG, average daily feed intake (ADFI), and FCR were analyzed.

### 2.5. Diarrhea Incidence

To determine the incidence of diarrhea, piglet feces were observed twice per day (9:00 a.m. and 3:00 p.m.) and classified into one of the following grades based on appearance; grade 1, hard cylinders; grade 2, soft cylinders; grade 3, thick and mushy feces; and grade 4, sloppy feces. In the present study, grade 4 was defined as diarrhea. The diarrhea incidence was the sum of piglets with diarrhea once or more throughout the experimental period, divided by the total number of piglets in each group.

### 2.6. Blood Sampling

Blood was collected from the jugular vein immediately after weighing on days 1, 7, 14, and 20. A 21-gauge needle (VENOJECT II; Terumo, Tokyo, Japan) was used to harvest blood for storage in 5-mL collection tubes containing EDTA-Na.

### 2.7. Blood Hematology Analysis

Blood hematology analyses, including white blood cell (WBC) count, red blood cell count, lymphocyte count, and neutrophil count, were performed using a pocH-100iV Diff hematology analyzer (Sysmex Corp., Kobe, Japan).

### 2.8. Plasma Collection and Immunoglobulin Analysis

After hematology analyses, the blood was centrifuged for 20 min (3000 rpm) at 4 °C to obtain plasma. The plasma was immediately subjected to biochemical analyses, and the remaining plasma was stored at −80 °C for subsequent use.

Immunoglobulin A (IgA) and immunoglobulin G (IgG) levels were measured by enzyme-linked immunosorbent assay kits (COSMOBIO, Tokyo, Japan).

### 2.9. Blood Biochemical Examination

Blood biochemical examinations were performed using an automatic dry-chemistry analyzer (DRI-CHEM 3500s; Fujifilm, Tokyo, Japan). The analysis included glutamic pyruvic transaminase (GPT), glutamic oxaloacetic transaminase (GOT), gamma glutamyl transferase (GGT), glucose, ammonia, amylase, and total protein.

### 2.10. Cortisol Measurement

The cortisol concentration in plasma was determined using duplicate enzyme-linked immunosorbent assay kits (Cayman Chemical Company, Ann Arbor, MI, USA) in accordance with the manufacturer’s protocol.

### 2.11. Actual and Relative Weights/Lengths of Organs and Intestines

After the feeding trial, two piglets near the average BW for each pen underwent induction of deep anesthesia via thiopental sodium (Ravonal 0.5 g; Mitsubishi Tanabe Pharma, Osaka, Japan) injection into the jugular vein; they were then sacrificed. Necropsies were performed and the organs (liver, pancreas, spleen, kidney, stomach, small intestine, large intestine, and thymus) were carefully removed. The weights of all organs, including individual intestinal tract sections, were measured. The relative organ weights were calculated as the organ weight divided by BW (g/kg). The lengths of individual intestinal tract sections were measured and the relative lengths of intestinal tract sections to piglet BW were also calculated (cm/kg). Specific parts of the small and large intestines were removed (described in next section) and fixed in 4% paraformaldehyde for the examination of intestinal morphology.

### 2.12. Intestinal Morphology

The following samples of intestinal tract were harvested immediately after measurement of the intestinal tract length: 2 cm of duodenum, 10 cm from the stomach; 2 cm of jejunum, 50 cm from the duodenal sampling site; 2 cm of ileum, 50 cm from the cecum; and 2 cm of colon, 50 cm from the cecum. The samples were rinsed in cold 0.1 M phosphate-buffered saline, then divided into two sections (1 cm each) and fixed in 4% paraformaldehyde in 0.1 M phosphate-buffered saline. The fixed tissues were then carefully oriented and embedded in Tissue-Tek OCT (Sakura Finetechnical Co., Ltd., Tokyo, Japan) and sectioned into 6-µm slices using a Leica CM1850 cryomicrotome (Leica Microsystems Co., Ltd., Wetzlar, Germany). The tissue sections were stained with hematoxylin and eosin for morphological analysis. Cross-sectional slices were viewed with an Olympus IX71 microscope (Olympus, Tokyo, Japan) and images were produced using DP Controller (version 1.2.1.108, Olympus, Tokyo, Japan). One cross-section with 10 consecutive intact, well-oriented crypt–villus units was selected from each 1-cm section of intestinal sample; because villi in the flat part between folds exhibited more regularity than did villi on the folds, only sections with flat parts were selected. The villus height and crypt depth were analyzed using the open-source software ImageJ (version 1.52 k, National Institutes of Health, Bethesda, MD, USA). Villus height was measured from the tip of the villus to the villus crypt junction, crypt depth was defined as the depth of the invagination between adjacent villi, and the ratio of villus height to crypt depth was calculated. Mucosal thickness was measured from the tip of the villus to the bottom of the muscularis mucosae.

### 2.13. Statistical Analysis

Data analysis was performed using JMP Pro software (version 15.2.0, SAS Institute Inc., Cary, NC, USA). One-way analysis of variance was used to compare differences among experimental groups. When the *p*-value from analysis of variance was <0.05, pairwise differences were assessed using the Tukey–Kramer honestly significant difference test. *p*-values < 0.05 were considered to indicate statistical significance. Results are presented as the means ± standard errors of the mean.

## 3. Results

### 3.1. Growth Performance

As shown in Table 5, 0.2% and 0.3% MGM-P supplementation did not influence growth performance indices, including the ADG, ADFI, or FCR.

### 3.2. Diarrhea Incidence

As shown in Figure 1, the diarrhea incidences were 12.5% and 12.5% in the control and 0.2% MGM groups, respectively, during the 20-day post-weaning period. However, no diarrhea was observed in the 0.3% MGM group.

### 3.3. Blood Hematology Analysis

As shown in Figure 2, there were large differences in WBC counts, particularly the neutrophil counts, between the treatment and control groups at 14 days post-weaning; there were no differences in red blood cell count or platelet count. The WBC and neutrophil counts were significantly lower in the 0.3% MGM group than in the control group (*p* < 0.05).

### 3.4. Blood Immunoglobulin Analysis

As shown in Figure 3, there were no changes in plasma IgA or IgG concentrations throughout the experimental period.

### 3.5. Blood Biochemical Analysis

As shown in Figure 4, MGM-P supplementation did not influence GPT or GOT concentrations in piglet plasma (*p* > 0.05). The concentrations of GGT were lower in the treatment groups than in the control group on day 20 post-weaning. Both glucose and ammonia concentrations tended to decrease in the treatment group; they were significantly lower in the 0.2% MGM group (*p* < 0.05) than in the control group on day 20 post-weaning. The amylase level exhibited a similar tendency, but the difference was not statistically significant (*p* > 0.05). The total protein level did not significantly differ among groups (*p* > 0.05).

### 3.6. Cortisol Analysis

As shown in Figure 5, the cortisol concentration on the day of weaning was significantly higher in the control group than in the treatment group (*p* < 0.05). Throughout the experimental period, the cortisol concentration in the 0.3% MGM group remained at a consistently low level.

### 3.7. Actual and Relative Weights/Lengths of Organs and Intestines

Pathological autopsy confirmed that no organ abnormalities were present. Table 6 shows the effects of MGM-P supplementation on organ weight or length and relative weight or relative length in weaned piglets. All results revealed no significant deviations between the treatment and control groups (*p* > 0.05).

### 3.8. Intestinal Morphology

Figure 6 shows that, compared with piglets in the control group, piglets in the treatment groups had more complete villus structures in all small intestine sampling sites (duodenum, jejunum, and ileum) on day 20 post-weaning; in particular, piglets treated with 0.3% MGM-P appeared to have denser villi. Morphometric measurements are summarized in Figure 7. In particular, piglets in the 0.3% MGM group displayed increased jejunal villus height (*p* < 0.05) and decreased ileal crypt depth (*p* < 0.05), compared with piglets in the control group. Moreover, thinner colonic mucosae were detected in the 0.3% MGM group.

## 4. Discussion

Whether tannins used as a feed additive have positive effects on early-weaned piglet growth is controversial. Several studies have indicated that tannins are an anti-nutrition factor that can reduce feed intake and nutrient digestibility (particularly protein), thus inhibiting piglet growth [17,18]. Lizardo et al. [19] reported a reduction in the growth performance of weaned piglets fed a tannin-rich sorghum diet. However, the present study showed that neither concentration of MGM-P (0.2% and 0.3%) had a negative impact on animal growth performance, ADFI, and FCR throughout the experiment. Few studies have investigated the effects of quebracho tannin on intestinal health and growth in weaned piglets. The heterogeneous responses seen among studies may be related to the dose of tannin, type of tannin product, age of the animals, ingredients of the basal diet, and hygiene and storage status.

Diarrhea is an important cause of death in early-weaned piglets, and causes extensive economic losses worldwide [20]. Early weaning frequently triggers intestinal inflammation and disrupts intestinal barrier function, thus rendering the intestinal structure vulnerable to bacterial invasion (which is an important cause of diarrhea in piglets) [21]. The present study showed that 0.3% MGM-P supplementation significantly reduced the incidence of diarrhea among early-weaned piglets by increasing jejunal villus height, decreasing ileal crypt depth, and reducing colonic mucosa. In some studies of intestinal inflammation, the WBC count dramatically increased soon after weaning [22,23]. In the present study, diarrhea gradually appeared in the control and 0.2% MGM groups after weaning, but no diarrhea was observed in the 0.3% MGM group. The rate of change in WBC count was lowest in the 0.3% MGM group, and showed significant group differences at 14 days post-weaning. Furthermore, the neutrophil count was significantly lower in the 0.3% MGM group than in the other groups. Neutrophils are closely associated with intestinal homeostasis and disease; they represent a key component of the innate response during an inflammatory reaction [24] and have important roles in the defense against bacterial and fungal pathogen invasion [25]. Yi et al. [26] reported that diarrhea in weaned piglets was accompanied by a substantial increase in neutrophil recruitment to the intestinal tract. The results of the present study are consistent with the above finding, where the prevention of diarrhea in weaned piglets was associated with low levels of neutrophils.

Liu et al. [27,28] demonstrated that the use of chestnut tannins as feed additives could increase immunoglobulin levels in pigs and broilers. However, the present study did not observe changes in immunoglobulin concentrations, suggesting that the results were related to tannin type; thus far, there have been few reports describing the effect of MGM-P supplementation on immunoglobulin levels in pigs.

Hydrolyzed tannins readily degrade in the gastrointestinal tract and enter the systemic circulation, causing liver toxicity. Filippich et al. [29] showed that after oral administration of >7.5 mg/g BW of crude extract from yellow-wood leaves, obvious liver lesions occurred within 48 h; a hepatotoxic component known as punicalagin (hydrolyzed tannin) was subsequently isolated. However, little is known regarding the toxicity of condensed tannins. Plasma GPT, GOT, and GGT are important indicators of liver injury. When the liver is damaged, the plasma levels of GPT, GOT, and GGT increase [30,31]. In the present study, no changes were observed in GPT or GOT; the concentration of GGT was lower in the 0.3% MGM group. These findings demonstrated that MGM-P addition was safe, and the piglets’ livers were in good condition. McDougall et al. [32] reported that tannin-rich extracts were effective amylase inhibitors. In the present study, 0.2% MGM-P supplementation led to decreased amylase activity and glucose concentration. The 0.3% MGM group exhibited similar tendencies, but the differences were not statistically significant. These results suggest that further investigations are needed to evaluate the relationships of amylase activity and glucose concentration with growth performance and feed intake, according to additive levels. Furthermore, there were no differences in total protein levels among groups, consistent with the lack of any negative impact on animal growth performance, ADFI, or FCR throughout the experiment. The ammonia concentration tended to be lower in the treatment than control group. The hepatic urea cycle is the main route for conversion of ammonia into blood urea nitrogen; therefore, the concentration of ammonia can serve as an indicator of liver function [33]. Liver impairment increases the ammonia concentration in the blood, which exacerbates systemic inflammation and liver injury [34]. MGM-P supplementation is beneficial for the liver. Tannins have been shown to exert a protective effect on the liver, presumably by enhancing resistance to oxidation and inflammation [35,36].

Cortisol is a glucocorticoid produced by the adrenal cortex under stress, and thus is widely used to detect physiological stress in pigs [37]. At the beginning of our experiment, the cortisol concentration was higher in the control group than the other groups; we presumed that this was due to interindividual differences in reactions to bleeding, although the animals were randomly assigned to groups. Throughout the experimental period, 0.3% MGM-P supplementation decreased plasma cortisol levels, suggesting improved physiological condition; this might explain why tannins did not negatively affect growth and diarrhea (i.e., because stress is closely related to growth performance) [2].

In the present study, detailed pathological examinations were performed during dissection, and revealed that 0.2% and 0.3% MGM-P had no pathological effects on the organs. These results are consistent with the findings of Wang et al. [38] that adding various plant extracts to the diet of 21-day-old weaned piglets had no pathological effects on organs after 14 days of supplementation. The intestinal development and digestive function of piglets are severely impaired during the weaning process [39]. Importantly, we found no changes in organ weight, relative organ weight, intestine length, or relative intestine length.

Integrity of the morphological structure is necessary to ensure good intestinal function; a complete mucosal structure is more protective against harmful substances. Weaning stress is associated with poorer intestinal barrier function [40]; it causes villous atrophy and crypt hyperplasia in the intestinal tracts of piglets, resulting in dysfunctional nutrient digestion and absorption [41]. Bilić-Šobot et al. [42] reported that supplementation with 3% hydrolyzable chestnut tannins significantly increased duodenal villus height in fattening boars. Our results were similar, in that 0.3% MGM-P significantly increased the jejunal villus height. Biagi et al. [10] found that ileal crypt depths tended to decrease in animals supplemented with tannins (2.25 or 4.5 g/kg). The present study showed that supplementation with MGM-P, particularly 0.3% MGM-P, led to noticeably shallower ileal crypts. These shallower crypts may help to reduce the severity of post-weaning diarrhea in piglets because of robust secretory functioning in the small intestine crypts [10]. The main functions of the colon are to absorb water and mineral salts from chyme. In this study, the piglets supplemented with 0.3% MGM-P had thinner colonic mucosa than those in the other two groups, which suggests that 0.3% MGM-P might have thinned the colonic mucosa, thus helping the piglets to absorb water from the stool and reducing the incidence of diarrhea.

## 5. Conclusions

The present study demonstrated that MGM-P supplementation helped prevent diarrhea in 21-day-old weaned piglets, particularly those in the 0.3% MGM group. This treatment positively influenced piglet health without adversely affecting growth performance. Practical studies of the mechanisms underlying the effects of MGM-P additives and the optimal amounts thereof are needed to confirm our findings.

## Figures and Tables

**Figure 1 animals-11-03316-f001:**
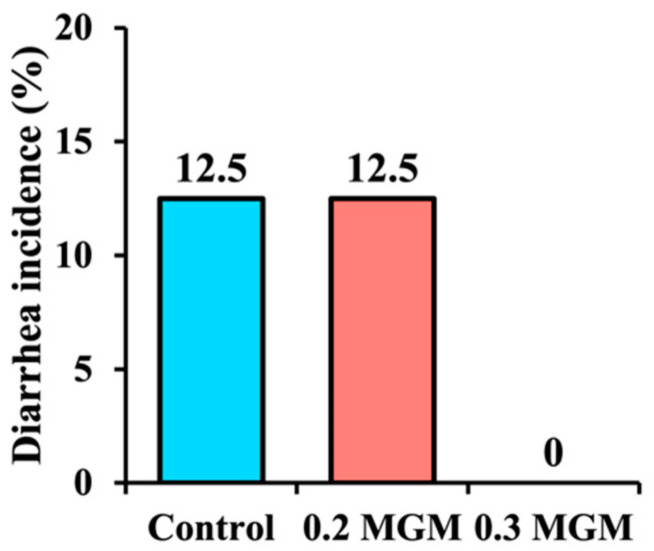
Effects of MGM-P supplementation on diarrhea incidence in weaned piglets.

**Figure 2 animals-11-03316-f002:**
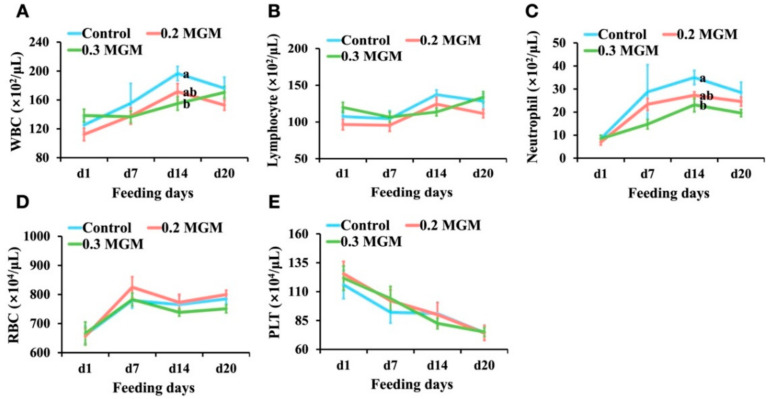
Effects of MGM-P supplementation on blood hematology parameters in weaned piglets. Blood was collected on d1 before the provision of feed with MGM-P on the day of weaning. Values are expressed as mean ± SEM; *n* = 8. Different lowercase letters on the right side of the nodes indicate significant differences (*p* < 0.05; Tukey HSD test). (**A**) the changes of the white blood cell count; (**B**) the changes of the lymphocyte count; (**C**) the changes of the neutrophil count; (**D**) the changes of the red blood cell count; (**E**), the changes of the platelet count.

**Figure 3 animals-11-03316-f003:**
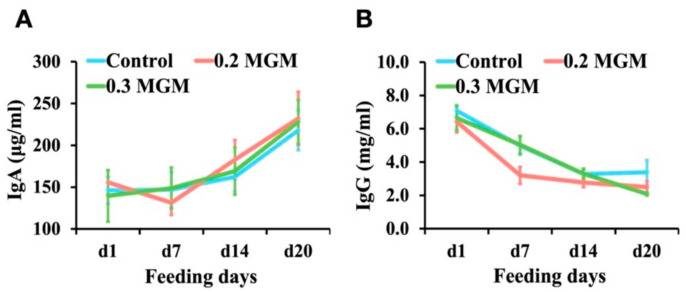
Effects of MGM-P supplementation on blood immunoglobulin levels in weaned piglets. Blood was collected on d1 before the provision of feed with MGM-P on the day of weaning. Values are expressed as mean ± SEM; *n* = 8. Different lowercase letters on the right side of the nodes indicate significant differences (*p* < 0.05; Tukey HSD test). (**A**) the changes of the immunoglobulin A concentration; (**B**) the changes of the immunoglobulin G concentration.

**Figure 4 animals-11-03316-f004:**
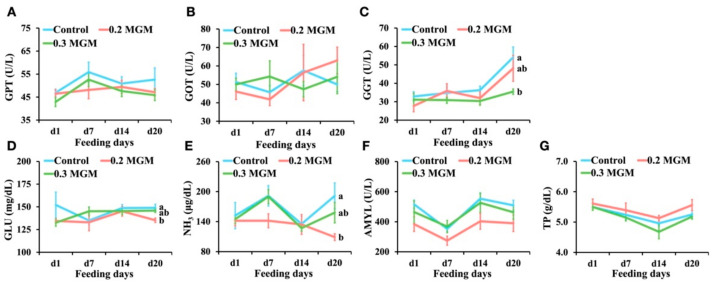
Effects of MGM-P supplementation on blood biochemical parameters of weaned piglets. Blood was collected on d1 before the provision of feed with MGM-P on the day of weaning. Values are expressed as mean ± SEM; *n* = 8. Different lowercase letters on the right side of the nodes indicate significant differences (*p* < 0.05; Tukey HSD test). (**A**) the changes of the glutamic pyruvic transaminase level; (**B**) the changes of the glutamic oxaloacetic transaminase level; (**C**) the changes of the gamma glutamyl transferase level; (**D**) the changes of the glucose concentration; (**E**) the changes of the ammonia concentration; (**F**) the changes of the amylase level; (**G**) the changes of the total protein concentration.

**Figure 5 animals-11-03316-f005:**
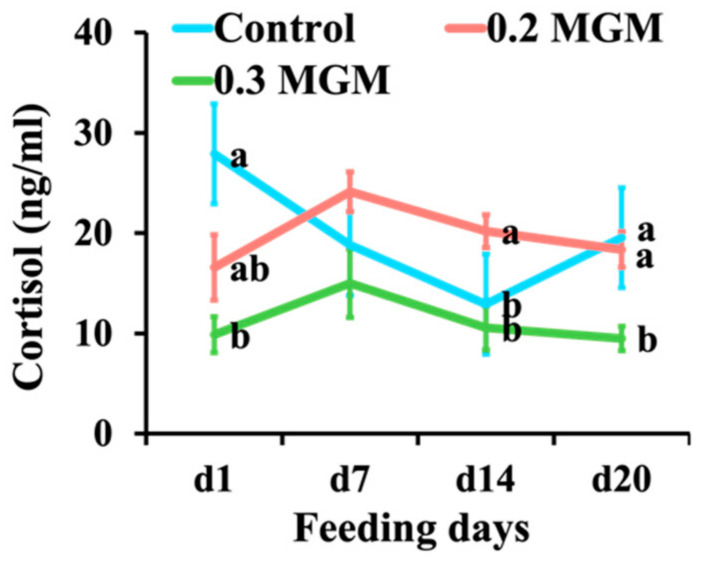
Effects of MGM-P supplementation on the blood cortisol concentration of weaned piglets. Blood was collected on d1 before the provision of feed with MGM-P on the day of weaning. Values are expressed as mean ± SEM; *n* = 8. Different lowercase letters on the right side of the nodes indicate significant differences (*p* < 0.05; Tukey HSD test).

**Figure 6 animals-11-03316-f006:**
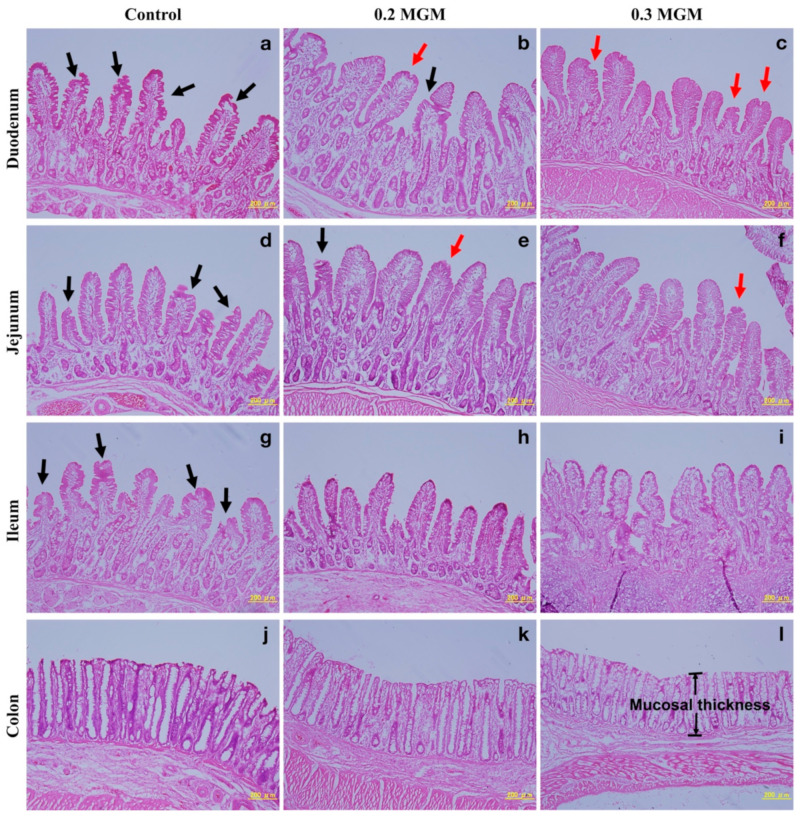
Representative histological micrographs of the duodenum, jejunum, ileum, and colon weaned piglets at 20 days post-weaning, produced by hematoxylin and eosin staining. Scale bar, 200 μm. Abbreviations: Control, control group; 0.2 MGM, 0.2% MGM-P group; 0.3 MGM, 0.3% MGM-P group. Pathological changes at the tips of intestinal villi during weaning are indicated by red (slight damage) and black (serious damage) arrows. The exposed lamina propria are clearly visible in the control group (**a**,**d**,**g**). The mucosal integrity is relatively high in the treatment group (**b**,**c**,**e**,**f**,**h**,**i**) especially in 0.3 MGM group (**f**,**h**,**i**). The mucosal thickness of the colon is relatively thick in the control group (**j**) and 0.2 MGM group (**k**) than that in the 0.3 MGM group (**l**) (including the muscularis mucosae).

**Figure 7 animals-11-03316-f007:**
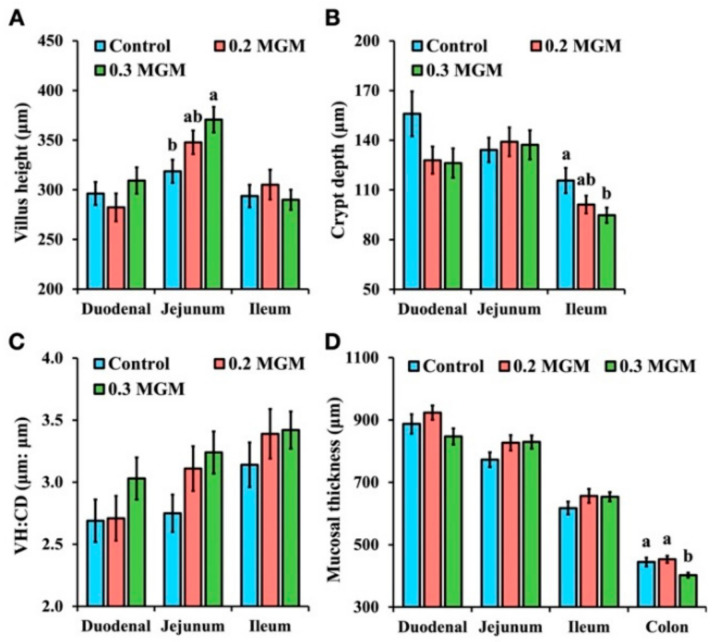
Effects of MGM-P supplementation on the intestinal morphology of weaned piglets. Abbreviations: VH:CD, villus height/crypt depth; Control, control group; 0.2 MGM, 0.2% MGM-P group; 0.3 MGM, 0.3% MGM-P group. Values are expressed as mean ± SEM; *n* = 4. Different lowercase letters above the bars indicate significant differences (*p* < 0.05; Tukey HSD test). (**A**) comparison of villus height in different groups (duodenum, jejunum and ileum); (**B**), comparison of crypt depth in different groups (duodenum, jejunum and ileum); (**C**) the ratio of villus height to crypt depth in different groups (duodenum, jejunum and ileum); (**D**) comparison of mucosal thickness in different groups (duodenum, jejunum, ileum and colon).

**Table 1 animals-11-03316-t001:** Technical specification of MGM-P.

Characteristic	Criterion
Polyphenols (mg catechin/g)	>500
Humidity (maximum)	15
pH	4.5 ± 1

**Table 2 animals-11-03316-t002:** Experimental animal allotment.

Group	Mean	*n*	SEM	*p*-Value	CV (%)
Control	6.56	8	0.33	0.99	0.58
0.2 MGM	6.49	8	0.28		
0.3 MGM	6.49	8	0.32		

Abbreviations: n, number of piglets; SEM, standard error of the mean; CV, coefficient of variation.

**Table 3 animals-11-03316-t003:** Ingredients and chemical composition of basal diet (as-fed basis) ^1^.

Ingredient	Content (%)
Corn	34.45
Defatted milk powder	18.00
Fatty powder	6.20
Sugar	10.00
Soybean meal	25.00
Fish meal	4.50
Calcium diphosphate	0.20
Calcium carbonate	0.65
Salt	0.20
B vitamins	0.15
Vitamins A, D and E	0.10
Trace minerals	0.15
L-lysine hydrochloride	0.06
DL-Methionine	0.09
L-Threonine	0.03
Copper sulphate	0.21
Vitamin K3	0.01
Total	100

^1^ The other diets were based on this diet, to which MGM-P was added in different proportions.

**Table 4 animals-11-03316-t004:** Chemical composition of basal diet.

Chemical Composition	Content (%)	Amino Acid	Content (%)
DM	90.50	Contained	
CP	22.60	Arginine	1.32
EE	6.60	Histidine	0.63
CF	1.10	Isoleucine	0.99
Ash	5.60	Leucine	2.03
NFE	54.60	Lysin	1.56
DE (Mcal/kg)	3.70	Methionine + cysteine	0.83
Ca	0.81	Phenylalanine + tyrosine	1.92
NpP	0.45	Threonine	0.96
Na	0.26	Tryptophan	0.28
Cl	0.36	Valine	1.15
K	0.99	Digestible	
Mg	0.14	Arginine	1.22
Fe (mg/kg)	182.18	Histidine	0.58
Zn (mg/kg)	105.32	Isoleucine	0.88
Mn (mg/kg)	87.51	Leucine	1.83
Cu (mg/kg)	125.29	Lysin	1.42
I (mg/kg)	1.95	Methionine + cysteine	0.74
Se (mg/kg)	0.30	Phenylalanine + tyrosine	1.47
Vitamin A (IU/kg)	100,051.62	Threonine	0.85
Vitamin D (IU/kg)	2000	Tryptophan	0.25
Vitamin E (IU/kg)	20.04	Valine	1.01
Vitamin K (IU/kg)	0.57		
Thiamine (mg/kg)	5.15		
Riboflavin (mg/kg)	15.38		
Pantothenic acid (mg/kg)	27.83		
Nicotinic acid (mg/kg)	25.63		
Vitamin B6 (mg/kg)	5.93		
Choline (mg/kg)	1204.8		
Vitamin B12 (μg/kg)	21.88		
Biotin (mg/kg)	0.16		
Folic acid (mg/kg)	0.36		

Abbreviations: DM, dry matter; CP, crude protein; EE, ether extract; CF, crude fiber; NFE, nitrogen free extract; DE, digestible energy.

**Table 5 animals-11-03316-t005:** Effects of MGM-P supplementation on growth performance in weaned piglets.

Item	Initial BW (kg)	Final BW (kg)	ADG (kg)	ADFI (kg)	FCR
Control	6.56 ± 0.33	17.56 ± 0.65	0.55 ± 0.02	0.82	1.49
0.2 MGM	6.49 ± 0.28	17.46 ± 0.93	0.55 ± 0.04	0.81	1.47
0.3 MGM	6.49 ± 0.32	17.79 ± 0.82	0.57 ± 0.03	0.84	1.47

Values of BW and ADG are expressed as mean ± SEM; *n* = 8. There were no statistically significant differences among the three groups based on one-way analysis of variance.

**Table 6 animals-11-03316-t006:** Effects of MGM-P supplementation on the organ weight/length of weaned piglets.

Measurement	Control	0.2 MGM	0.3 MGM
Organ weight/length			
Liver (g)	425.28 ± 39.64	478.38 ± 45.16	474.08 ± 31.26
Pancreas (g)	31.45 ± 4.52	35.70 ± 2.78	34.73 ± 2.19
Spleen (g)	32.40 ± 2.26	35.65 ± 2.26	35.08 ± 2.93
Kidney (g)	97.90 ± 5.19	115.28 ± 8.31	103.50 ± 6.45
Stomach (g)	76.58 ± 6.28	78.98 ± 7.53	78.85 ± 11.64
Small intestine weight (g)	598.73 ± 34.20	804.40 ± 85.75	590.50 ± 39.51
Small intestine length (m)	11.14 ± 0.40	12.68 ± 0.47	11.93 ± 0.79
Large intestine weight (g)	189.75 ± 37.24	230.03 ± 19.61	187.63 ± 16.50
Large intestine length (m)	1.98 ± 0.37	2.58 ± 0.02	2.61 ± 0.31
Thymus (g)	35.73 ± 1.33	40.83 ± 11.81	41.43 ± 8.93
Relative organ weight/length			
Liver (%)	2.35 ± 0.10	2.58 ± 0.18	2.60 ± 0.16
Pancreas (%)	0.18 ± 0.02	0.19 ± 0.01	0.19 ± 0.01
Spleen (%)	0.18 ± 0.02	0.19 ± 0.01	0.19 ± 0.01
Kidney (%)	0.55 ± 0.01	0.62 ± 0.02	0.57 ± 0.02
Stomach (%)	0.43 ± 0.02	0.43 ± 0.03	0.42 ± 0.03
Small intestine weight (%)	3.33 ± 0.09	4.37 ± 0.48	3.24 ± 0.18
Small intestine length (cm/kg)	62.45 ± 4.33	68.63 ± 2.33	65.96 ± 5.80
Large intestine weight (%)	1.04 ± 0.15	1.24 ± 0.07	1.02 ± 0.05
Large intestine length (cm/kg)	11.04 ± 2.08	14.00 ± 0.58	14.36 ± 1.62
Thymus (%)	0.20 ± 0.01	0.22 ± 0.05	0.22 ± 0.04

All data are expressed as mean ± SEM; *n* = 4. There were no statistically significant differences among the three groups based on one-way analysis of variance.

## Data Availability

The data that support the findings of this study are available from the corresponding author upon reasonable request.

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
