# Peer review of "Effects of Supplementation with a Quebracho Tannin Product as an Alternative to Antibiotics on Growth Performance, Diarrhea, and Overall Health in Early-Weaned Piglets"

_animals, 2021, doi:10.3390/ani11113316_

Round 1

Reviewer 1 Report

The manuscript, # animals-1452362 by Min Ma et al. assessed the feasibility of using a quebracho extract tannin product as an alternative antibiotic for weaned piglets by investigating its effects on growth performance, diarrhea and overall health. Their results showed that the quebracho extract tannin product could prevent diarrhea occurring and positively affect piglet health without any adverse effects on growth performance. However, due to the following concerns, I think it can be considered for publication after the following major revisions.

Major comments

-This research was supported financially by Kawamura Ltd and the commercial MGM-P was provided by Kawamura Ltd., also. In the reviewer’s opinion, this manuscript is more like an advertisement article from a commercial company. How the authors could avoid biased manuscript to the quebracho extract tannin product?

-The objective of this study was investigated the effects of supplementation with a quebracho extract tannin product as an alternative antibiotic on growth performance, diarrhea, and overall health in early weaned piglets. However, these was no an antibiotic treatment as positive control? How the experiment results could prove the hypothesis?

-Authors need to show the data of Figure1, 2, 3, 4, and 5 by tables, if you can.

Minor comments

Line 32: Double check the sentence of “And positively affects piglet health without any adverse effects on growth performance.”

Line 55-58: Double check the sentence of “However, this early weaning carries substantial physiological, environmental, and social challenges for piglets, including abrupt separation from their mothers, exposure to unfamiliar piglets, establishment of a new social hierarchy, modified housing, and changes in both food sources and drinking water.”

Line 101-102: Authors stated that “tannins are present in several feedstuffs and ingredients for animal nutrition, such as corn, wheat and barley”. Did the authors measure the concentration of tannin in the basal diets? How to explain the effects of tannin in the basal diets on the experiment results?

Line 139-158: Regarding the diet, Table 2 should show each ingredient in the diet, the data of ingredients in current table 2 is not clear yet. Authors need to list each ingredient of “animal feed”, “cereals” and “others” as the note of “Abbreviations” in table 2. And usually, DE or ME of the feed is needed in the table. Authors need to add it, as well as amino acid content, calcium and phosphorus content, digestible amino acids content.

Line 199: What was the abbreviation “ASF”? Please added it.

Line 122-123: The authors stated that “24 piglets were then selected and divided into three groups (n = 8 per group) according to weight and sex”. However, the individual piglet was regarded as an experimental unit in statistical analysis. This is incomprehensibility. Statistical analysis must be re-implemented.

Line 330-331: Double check the sentence of “…the dose of the tannin supple-mentation…”.

Line 340-341: Double check the sentence of “Withe onset of intestinal inflammation, the WBC count often increases dramatically soon after weaning”.

Line 410-413: Double check the sentence of “…which is suggesting that 0.3% MGM also by to thin colonic mucosa helped the…”.

Line 437: Please check the format of Reference. The abbreviation of some journals was all in capitals, but the abbreviation of others journals only capitalized the first letter. Moreover, the title of these references should be just capitalized the first word. Please revised them following the author guidelines.

Author Response

We thank reviewer 1 for the valuable comments and respond to them below. We hope that these responses are satisfactory.

Answer

First, this study was performed pursuant to a contract between the faculty of the Graduate School of Agricultural and Life Sciences, University of Tokyo and Kawamura Ltd. All results were submitted to both entities. Therefore, the study was performed according to the guidelines of the faculty of the Graduate School of Agricultural and Life Sciences, University of Tokyo.

Second, all of the authors, the individual in charge of the experiment, and the individual in charge of the facility were required to comply with University of Tokyo Animal Experiment Implementation Regulations and University of Tokyo Animal Experiment Implementation Manual.

Third, all authors declare that this study was performed independently and only the results were submitted to Kawamura Ltd. without any prior discussion. All authors certify that there is no conflict of interest involving Kawamura Ltd.

Fourth, we are conducting additional investigations. We plan to publish the findings soon.

Answer

When planning the present study, we could not purchase the in-feed antibiotic in Japan because the company that provides it did not have sufficient inventory. Thus, the present study focused on preventing diarrhea, preserving overall health, and assessment of potential negative effects on growth performance in early weaned piglets after supplementation with the quebracho tannin product. In a future experiment, we plan to adjust the level of quebracho tannin and will establish a positive control group by purchasing in-feed antibiotics from an international supplier.

Answer

Because there are four stages of change, we feel that a graph aids clarity.

 Minor comments

Answer: The text has been modified as suggested.

Answer: The concentration of tannin (i.e., tannin acid) in the basal diet was measured by Japan Food Research Laboratories. Because both the control and treatment groups used the same basal diet, we have concluded that the different results were related to the treatments.

Answer: In accordance with the reviewer’s comment, we changed Table 1 to Table 2 and listed each ingredient; we also provided a new Table 4 to show the chemical composition of the basal diet. The DE content is listed in Table 4.

Answer: ASF represents “antibacterial substance-free.”

Answer: In accordance with the reviewer’s comment, we have removed the following text from Line 228: “An individual piglet was regarded as an experimental unit.”

Answer: The text has been modified to read “dose of tannin”.

Answer: The text has been modified to “with one set”

Answer: The text has been modified, i.e., “also” has been changed to “might”.

Answer: We have modified the Reference list to ensure that it adheres to the author guidelines.

Reviewer 2 Report

The work of Ma and collaborators want to exploit the use of the quebracho extract tannin in weaned piglets as a possible alternative to antibiotics.

Strengths

The topic of this study is very interesting and actual, especially the determination of the correct dosage of tannins to maximize the positive effects. The authors try two different concentrations of quebracho tannins (0,2% and 0,3%); The addition of 0,3% of quebracho tannins reduces diarrhea remarkably in weaned piglets.

Limitations

The major limitation is the number of animals (8 per group), especially regarding the growth performance and diarrhea incidence. The addition of 0,3% decrease zero the diarrhea incidence, an addition of 0,2% doesn’t give any effects. The authors can explain (or have an idea) as a minimal difference of concentrations gives a strong difference?

INTRODUCTION

The introduction is generally adequate. The authors can increase and explain better the difference between hydrolyzable and condensed tannins. The difference is important for the difference in the performance and more…

MATERIALS AND METHODS

Materials and methods are adequate.

The authors used a commercial product. Please, it’s possible to characterize better the product? Extraction methods? Composition?

Blood analysis: how many animals do you use?  All animals?

RESULTS and DISCUSSION

The results are clear, but the low number of animals analyzed and the ideal growing conditions can emphasize the results. This work must be considered ONLY a preliminary study; the same authors recognize the need for more in-depth study in vivo and in vitro. 

MINOR REVISIONS

Line 28: please add a comma before and (anti-inflammatory activity, and …)

Line 29: Knowlogy -knowledge

Line 33: effectiveness-effective

Line 37: please add a comma before and (diarrhea, and overall..)

Line 66: antibiotics are

Line 126: without with any addition

Line 181: was stored at -80 °C to until next use

Line 208-209: 50 cm from the cecum is double

Line 243: 12,5% and 12,5% in the control and 0,2%MGM groups

Line 325: inhibiting pig’s growth

Line 340: White-whit

Line 356: and there is few report described effect- and there is few reportsdescribing the…

Line 367: these results suggesting suggest

Author Response

We thank reviewer 2 for the valuable comments, which we respond to below. We hope that these responses are satisfactory.

Answer: We thank the reviewer for this comment.

Answer: We agree that the number of animals included is a limitation. The publication of this study will help support our efforts to obtain additional funding for a larger number of animals. The present results have demonstrated that quebracho tannin can serve as an alternative to antibiotics in early weaned piglets; it protects against diarrhea and preserves overall health. As shown in Figure 2, the 0.2% MGM group also tended to show smaller increases in WBC and neutrophil counts within the first week after weaning. These findings suggest a dose-dependent effect.

Answer: The requested explanation has been added to Lines 92–100. Hydrolyzable tannins are hydrolyzed by weak bases, weak acids, or weak enzymes to produce carbohydrates and phenolic, gallic, and ellagic acids. Thus, hydrolyzable tannins are readily affected by basal diet composition. Hydrolyzed phenolic, gallic, and ellagic acids have potential antibacterial effects, which may to lead to variable research results. Condensed tannins have a stable structure and are not hydrolyzed. Condensed tannins bind to and precipitate proteins and various other organic compounds (e.g., amino acids and alkaloids), supporting their use in the treatment of weaned piglets.

Answer: The MGM-P technical specifications are provided in Lines 120–123. The manufacturer states that the extraction process should be conducted with hot water alone.

Answer: Yes, blood was analyzed for all animals.

RESULTS and DISCUSSION

Thanks. Yes, we have begun planning the next study.

Answer: The text has been modified as suggested.

 Answer: The text has been modified as suggested.

Answer: The text has been modified as suggested.

 Answer: The text has been modified as suggested.

 Answer: The text has been modified as suggested.

 Answer: The text has been modified as suggested.

 Answer: The text has been modified as suggested.

Answer: This indicates the locations of different samples: one location is the ileum and the other is the colon.

Answer: We apologize that we cannot locate the text “12.5% and 12.5% in the control and 0.2% MGM groups”. We respectfully request clarification from the reviewer.

Answer: The text has been modified as suggested.

Answer: The text has been modified as suggested.

Answer: The text has been modified as suggested.

Answer: The text has been modified as suggested.

Reviewer 3 Report

Dear authors,

Thank you for your contribution to the knowledge of the use of quebracho tannin as an alternative to antibiotics in weaned piglets.

I have some general comments and some more specific points.

In general, there seem to be some linguistic inaccuracies in the text: e.g. in the abstract where you say “This study assessed the feasibility of using a vegetable extract, MGM-P (quebracho tannin product) as an alternative antibiotics for weaned piglets”, it is not clear. Does It mean as an alternative TO antibiotics or are you saying that MGM-P are alternative antibiotics (they are not antibiotics!).

So, although I am not a native speaker, I recommend a linguistic review of the English text

Even if you have used only 3 sows for piglets production, it is generally preferable to form the experimental groups taking maternity into account in addition to sex and live weight.

You used single piglet ad experimental unit. With a little effort (27 animals vs 20) you could have also 3 pens for each treatment and the possibility to consider also the pen as experimental unit. 

In discussion, line 358. Whereas a positive effect of tannin on liver function is reported below, it would be better here to indicate the doses of liver toxicity of tannin reported in the literature for livestock.

Some minor points

Line 58. Please consider to use feed instead of food

Line 249 - As shown in Figure 2, there were

From line 329 to line 331 - Because only a few application studies of quebracho tannin took place  in the intestinal health and growth of post-weaning piglet. The heterogeneous responses  between the different studies probably related to the dose of the tannin supplementation. The 2 sentences are disconnected.

Author Response

We thank reviewer 3 for the valuable comments, which we have respond to below. We hope that these responses are satisfactory.

Answer: The revised manuscript has been carefully reviewed by an experienced editor whose first language is English and who specializes in editing papers written by researchers whose native language is not English.

Answer: As suggested, we will use this approach in future experiments.

Answer: As suggested, we will use this approach in future experiments.

 Answer: We have mentioned the liver toxicity of tannins for livestock according to previous literature.

Answer: The text has been modified as suggested.

 Answer: The text has been modified as suggested.

Answer: Sorghum tannins are also known as condensed tannins, but the additive results are distinct from quebracho tannin. Here, we wanted to explain why condensed tannins might have had different results.

Round 2

Reviewer 1 Report

The current manuscript , #1452362-peer-review-v2, is acceptable.

Reviewer 2 Report

Thanks for your reply.